**Data Availability Statement:** All relevant data are within the manuscript and its Supporting Information files.

# Development and validation of a prognostic tool: Pulmonary embolism short-term clinical outcomes risk estimation (PE-SCORE)

Anthony J. Weekes[1]*, Jaron D. Raper[1¤a], Kathryn Lupez[1¤b], Alyssa M. Thomas[1¤c], Carly A. Cox[1¤d], Dasia Esener[2], Jeremy S. Boyd[3], Jason T. Nomura[4], Jillian Davison[5], Patrick M. Ockerse[6], Stephen Leech[5], Jakea Johnson[3], Eric Abrams[2], Kathleen Murphy[4], Christopher Kelly[6], H. James Norton[7]

1 Department of Emergency Medicine, Atrium Health's Carolinas Medical Center, Charlotte, NC, United States of America, 2 Department of Emergency Medicine, Kaiser Permanente, San Diego, CA, United States of America, 3 Department of Emergency Medicine, Vanderbilt University Medical Center, Nashville, TN, United States of America, 4 Department of Emergency Medicine, Christiana Care, Newark, DE, United States of America, 5 Department of Emergency Medicine, Orlando Health, Orlando, FL, United States of America, 6 Division of Emergency Medicine, University of Utah Health, Salt Lake City, UT, United States of America, 7 Professor Emeritus of Biostatistics, Atrium Health's Carolinas Medical Center, Charlotte, NC, United States of America

¤a Current address: Department of Emergency Medicine, University of Alabama at Birmingham, Birmingham, Alabama, United States of America
¤b Current address: Department of Emergency Medicine, Tufts Medical Center, Boston, Massachusetts, United States of America
¤c Current address: Emergency Department, Houston Methodist Baytown Hospital, Houston, Texas, United States of America
¤d Current address: Emergency Medicine of Idaho, Meridian, Idaho, United States of America
* anthony.weekes@atriumhealth.org

## Abstract

### Objective

Develop and validate a prognostic model for clinical deterioration or death within days of pulmonary embolism (PE) diagnosis using point-of-care criteria.

### Methods

We used prospective registry data from six emergency departments. The primary composite outcome was death or deterioration (respiratory failure, cardiac arrest, new dysrhythmia, sustained hypotension, and rescue reperfusion intervention) within 5 days. Candidate predictors included laboratory and imaging right ventricle (RV) assessments. The prognostic model was developed from 935 PE patients. Univariable analysis of 138 candidate variables was followed by penalized and standard logistic regression on 26 retained variables, and then tested with a validation database (N = 801).

### Results

Logistic regression yielded a nine-variable model, then simplified to a nine-point tool (PE-SCORE): one point each for abnormal RV by echocardiography, abnormal RV by computed tomography, systolic blood pressure < 100 mmHg, dysrhythmia, suspected/confirmed

**Funding:** This project was supported by grant number R01HS025979 to AJW from the Agency for Healthcare Research and Quality (ahrq.gov). The content is solely the responsibility of the authors and does not necessarily represent the official views of the Agency for Healthcare Research and Quality. The funder had no role in study design, data collection and analysis, decision to publish, or preparation of the manuscript.

**Competing interests:** The authors have declared that no competing interests exist.

systemic infection, syncope, medico-social admission reason, abnormal heart rate, and two points for creatinine greater than 2.0 mg/dL. In the development database, 22.4% had the primary outcome. Prognostic accuracy of logistic regression model versus PE-SCORE model: 0.83 (0.80, 0.86) vs. 0.78 (0.75, 0.82) using area under the curve (AUC) and 0.61 (0.57, 0.64) vs. 0.50 (0.39, 0.60) using precision-recall curve (AUCpr). In the validation database, 26.6% had the primary outcome. PE-SCORE had AUC 0.77 (0.73, 0.81) and AUCpr 0.63 (0.43, 0.81). As points increased, outcome proportions increased: a score of zero had 2% outcome, whereas scores of six and above had $\geq$ 69.6% outcomes. In the validation dataset, PE-SCORE zero had 8% outcome [no deaths], whereas all patients with PE-SCORE of six and above had the primary outcome.

## Conclusions

PE-SCORE model identifies PE patients at low- and high-risk for deterioration and may help guide decisions about early outpatient management versus need for hospital-based monitoring.

## Introduction

An important indicator of acute pulmonary embolism (PE) of moderate to high severity is an acute increase in right ventricular pressure or size or decreased systolic function. PE-provoked right ventricle (RV) abnormality is commonly assessed in two ways: 1) laboratory surrogates of myocardial stretch and injury, and 2) imaging assessments for RV dilatation, pressure increases, and decreased systolic function. The most common diagnostic tests are natriuretic peptide and troponin, and imaging by computed tomography (CT) and echocardiography. Assessments for abnormal RV (abnlRV) are absent in validated clinical prognostic models, such as the original and simplified Pulmonary Embolism Severity Index (PESI and sPESI) and Hestia [1–3]. These prognostic prediction models utilized a limited set of candidate variables without pertinent imaging and laboratory measurements [4]. Risk of early clinical deterioration from worsening RV function is not captured in current prediction models [5–7].

The newer anticoagulants offer efficacy and safety in PE treatment, yet there is hesitancy to discharge those with acute PE. Hospitalization for PE is as high as 90%–95% in the U.S. and Europe, yet 41%–51% of PE patients are classified as low-risk by existing clinical prediction models [8–12]. Clinical algorithms, checklists, and prognostic models are being developed and updated to optimize the safety of outpatient management, improve prognostic accuracy for outcome(s), and provide guidance to reduce practice variation. Incorporation of imaging and laboratory assessments for PE-provoked abnlRV have now been incorporated into hybrid clinical algorithms [1, 7, 13–16], and some meta-analyses now support use of one or multiple RV assessment methods [4, 17, 18]. A consistent definition of PE-provoked abnlRV, however, is lacking [19–22].

Acute care providers are thus challenged to identify PE patients who are considered low-risk (and safe for early discharge) and those at greater risk of clinical deterioration without a clear guideline on RV assessment in acute PE. Providers must make disposition decisions driven by concerns for acute deterioration (respiratory failure, cardiac arrest, new dysrhythmia, sustained hypotension, and rescue reperfusion intervention) within the first days of PE diagnosis rather than events at 30 days or later. Thus, we aimed to develop and validate a

prediction model for the probability of deterioration or death within days of acute PE diagnosis in acute care settings.

## Materials and methods

### Study design and setting

This was a prospective, observational, multicenter study using two registry databases. The first database was the Pulmonary Embolism Short-term Clinical Outcomes Registry (PESCOR; clinicaltrials.gov NCT02883491), a prospective registry of patients who presented to six urban, academic, emergency departments (EDs) in the following locations during the pilot: San Diego, California; Newark, Delaware; Orlando, Florida; Charlotte, North Carolina; Nashville, Tennessee; and Salt Lake City, Utah. The cohort was chosen to allow for broad generalizability. By enrolling patients from a diverse set of EDs with geographic spread, we expected to capture the full spectrum of demographics and acute PE severity at presentation. The second registry was created after federal funding was secured for development of the prediction model (Short-term Clinical Deterioration After Acute Pulmonary Embolism; clinicaltrials.gov NCT03915925). The unfunded initial registry (PESCOR) was used for the validation. Both registries were populated by the same six EDs and had similar variables, data recording instruments, and outcome variables.

The development database was prospectively accrued between September 18, 2018 and December 14, 2020. The validation database was built between August 2016 to March 2019. The central site (located in Charlotte, North Carolina) prospectively enrolled consecutive patients; the other five sites prospectively enrolled on a convenience basis. During the early stages of the unfunded registry, the central site enrolled patients with written informed consent until its institutional review board (Atrium Health IRB) approved waiver of informed consent. The other five sites enrolled with written informed consent with approval from each of their institutional review boards. Once federal funding was secured, all sites used the central IRB (Advarra IRB), which approved the study protocol and waiver of written informed consent for enrollments at all sites (Advarra approval code PRO-00029256). The reporting of results adheres to the Transparent Reporting of a multivariable prediction model for Individual Prognosis or Diagnosis (TRIPOD) reporting criteria [23, 24].

### Participants

Inclusion and exclusion criteria were the same for both development and validation databases. Men and women 18 years or older with image-confirmed acute PE diagnosed within 12 hours of ED presentation were eligible for enrollment. Patients were excluded for any of the following reasons: age 17 years old and younger at the time of screening; refusal to participate in study; radiologist's determination that filling defects were chronic, resolving, or unchanged after comparison to previous CT, if available; empiric anticoagulation or escalated intervention initiated more than 12 hours before PE diagnosis; incidental identification of either segmental or subsegmental intraluminal filling defects on CT or unrelated to primary diagnostic workup or ED presentation.

### Data collection

The electronic case report included over 400 variable entry fields for prognostic model testing and other aims of the registry. For the prognostic tool, we collected 138 data elements on each patient, including vital signs at presentation, risk factors for PE, comorbidities, contemporaneous measurements of cardiac biomarkers [troponin and brain natriuretic peptide (BNP)], and

CT and goal-directed echocardiography evaluations performed early in ED management of the index PE event.

## Outcome measures

The primary composite outcome had morbidity and mortality outcomes of interest to emergency providers, which require hospital-based monitoring or time-sensitive interventions. We used a composite of death (all cause and PE-related) and clinical deterioration within five days of index PE confirmation. We incorporated and adapted a composite primary outcome previously used by researchers and considered to be important to providers and pulmonary embolism response teams in the USA and other countries [5, 7, 13, 25–28]. The individual components of the composite outcome have previously been reported on [5, 27]. Deaths were classified as PE-related when the site investigator reviewed the case and determined death was not likely to be due to another cause, such as septic shock or acute myocardial infarction. Elements of clinical deterioration included respiratory failure, cardiac arrest, new dysrhythmia, sustained hypotension requiring intravenous volume expansion or adrenergic medication, and rescue reperfusion intervention.

Respiratory failure was defined as respiratory distress associated with emergent interventions with mechanical ventilation (intubation, non-invasive positive pressure ventilation, or surgical cricothyrotomy). Cardiac arrest was defined as any unstable cardiac rhythm or absent electrical activity requiring cardiopulmonary resuscitation or advanced cardiac life support for asystole, pulseless electrical activity, ventricular fibrillation, or unstable ventricular tachycardia. New dysrhythmia was defined as the identification of atrial fibrillation with rapid ventricular response, atrial flutter, supraventricular tachycardia, stable ventricular tachycardia, or bradycardia that was not evident at ED presentation. Hypotension was defined as systolic blood pressure less than 90 mmHg (or a 40 mmHg decrease from baseline) or shock index >1 associated with administration of greater than 500 mL of intravenous fluids within 15 minutes for volume expansion or administration of norepinephrine, dopamine, or epinephrine infusion.

Major bleeding was attributed to treatment with anticoagulation or thrombolysis and not as a primary outcome of clinical deterioration due to PE severity. The presence of death or any clinical deterioration element within five days of hospitalization was considered to be positive for the primary outcome. The absence of death or clinical deterioration within five days post-PE confirmation was considered negative for primary outcome. Each patient could have more than one element of clinical deterioration.

Although not the focus of this report, our secondary outcome included the same events in 5 days with the addition of major bleeding, recurrence of venous thromboembolism (VTE), or subsequent hospitalization within 30 days.

## Predictor variables

We considered 138 candidate variables available at the point-of-care, including laboratory and imaging tests relevant to assessment of abnlRV, and those previously vetted by PE registries, sPESI, Hestia, and European Society of Cardiology (ESC) [3, 4, 15, 29, 30]. Predictor variables were measured and assessed while blinded to outcomes. We included symptoms, signs, and findings likely to represent higher PE severity. As an example, we chose syncope instead of shortness of breath or chest pain based on clinical experience and evidence in the literature [31–34]. We added a variable that factored in initial heart rate < 50 or > 100 bpm [35]. We included a component of Hestia that employed provider gestalt of medical and social support reasons for hospitalization as social determinants of health. Variables that addressed the safety

risk of PE treatment (including predispositions to bleeding) were not included as candidate variables for the primary outcomes of clinical deterioration. We report on the missingness of variables in the final prognostic model and associated outcomes of those with missing variable responses.

## Definitions of key predictor variables

**Echocardiography.** Goal-directed echocardiography (GDE) was performed by emergency physicians within four hours of PE confirmation with qualitative interpretation by site investigators. We reported the training level of providers performing the GDE (i.e., residency year, fellowship, or attending). We defined RV anatomy and physiology in PE using previously reported interpretation guidelines for abnormal [36–38]. We used a criteria grading system with high inter-rater reliability (kappa = 0.84) for severe RV dilatation between emergency physicians and cardiologists [37]. Quality assurance reviews were performed by experienced clinical ultrasound leaders.

Severe RV dilatation was defined as RV:LV basal diameter $\geq 1.0$ or basal RV diameter $> 42$ mm with blunting of the RV apex on two or more different windows. Severe RV systolic dysfunction was defined as a visual estimate of tricuspid annular planar systolic excursion (TAPSE) being 10 mm or less <u>and</u> RV free wall hypokinesis [38]. GDE was also assessed for flattening or deviation of interventricular septum (IVS) towards the left ventricle. The GDE score for PE-provoked RV dysfunction was assigned scores of zero to three. RV dilatation was considered to be a requirement for visual identification of PE-provoked RV dysfunction. The absence of RV dilatation was scored as zero, whereas one point each was assigned for RV dilatation, septal flattening or leftward deviation, and RV systolic dysfunction. When GDE was considered abnormal (scores of 1 to 3), a determination of whether the RV abnormality was acute, chronic, or indeterminate was included. We determined RV abnormality to be chronic based on the presence of RV free wall thickness $\geq 7$ mm or accompanying signs of LV abnormalities or previous echocardiography records reporting pre-existing RV abnormalities. We also noted if GDE image quality was inadequate for interpretation.

**Cardiac biomarkers.** Serum measurements were obtained within six hours of PE diagnosis to test for myocardial stretch and injury. For myocardial stretch, we used BNP with i-STAT BNP test cartridge (Abbott Point of Care, Abbott Park, IL), with a cut-off value of $> 90$ pg/mL. For sites that used N terminal BNP, the threshold cut-off value was 500 pg/mL. For myocardial injury, we used troponin i-STAT cTnI test cartridges (Abbott Point of Care, Abbott Park, IL), with cut-off value $\geq 0.07$ ng/mL. *In December 2019, the central site had an institution-wide replacement of point-of-care troponin I with high-sensitivity troponin, for which we used cut-off values of 20 ng/L for males and 12 ng/L for females.* We created binary categorical variables for natriuretic peptide and troponin elevation.

**Computed tomography pulmonary angiography (CTPA).** CTPA images were reviewed by board-certified radiologists unaffiliated with the research and blinded to clinical condition of patients. Using transverse 1 mm CT cuts, RV:LV basal diameter $\geq 1.0$ was considered indicative of RV dilatation. The proximal location of thrombus on CTPA was also reported (saddle, proximal portion of main right or left pulmonary arteries, lobar, segmental or subsegmental).

## Abstractor training

Before the study started, the principal investigator (PI) led detailed in-person discussions with site investigators to clearly define variables with field notes in REDCap case report forms. Monthly communication updates, central site monitoring, and in-person training sessions at national conferences were all employed for data cleaning. Before enrollment ended, the central

site performed univariable analyses to determine completeness and sensibility of entries. Verification queries were performed, with corrections made if necessary.

## Sample size

We used Peduzzi's rule for logistic regression to guide determination of sample size [39]. This rule declares the maximum number of independent (predictor) variables is no more than N/10, where N is the number of observations (subjects) in the smaller of the two groups (outcome dichotomous yes/no). We were prepared to accommodate up to 22 final variables. So, 220 subjects (220/10 = 22) were needed in the smaller (clinical deterioration yes) subgroup. Using an estimated 25% occurrence of clinical deterioration within several days (based on previously cited literature), sample size of 880 was required for the development database [6, 27, 40].

## Missing values

We reported the percentage of missing observations for each variable. Missing categorical data were marked as absent [41].

## Statistical analysis methods

**Data cleaning.** We performed three interim data cleans during the enrollment phase before importing to SAS for the final data clean after the final enrollment. During the enrollment phase, important variables were assessed for missingness and discrepancies during data cleaning. For example, we reported outliers in vital signs or laboratory measurement values to the site investigators. At the close of enrollments, descriptive statistics were used to examine predictor and outcome variables for sensibility and missingness. Instructions for corrective actions were assigned to the site investigator and clinical research team by referring to source documents within the electronic health records. After missingness was mitigated and sensibility of data optimized, the database was used for analysis. We then imported the development and external validation databases to SAS Enterprise Guide 7.1 (SAS Institute Inc., Cary, NC, USA).

We computed overall descriptive statistics on all variables in each dataset. We reported on the number of non-missing and missing values, the mean, median, standard deviation, minimum and maximum values for continuous variables. We used frequencies and percentages of each value (including missing values) for categorical variables. The PI and biostatistician inspected reports and made queries to verify and correct data as needed.

We created additional dichotomous variables using previously established cut-offs in validated prediction models or clinical guidelines (e.g., age > 80, systolic blood pressure < 100 mmHg, heart rate < 50 or > 100 bpm, initial oxygen saturation < 90%).

**Model development.** Fig 1 shows the steps taken to derive the prognostic model. We screened 138 candidate variables with bivariate analyses of the primary outcome in the development dataset. We used Student's t-test for continuous variables, Cochran-Armitage test for trend for ordinal variables, and the chi-square test for categorical variables. We chose a significance level of 5% <u>or</u> clinical importance as preliminary screening criteria for the full model testing and filtering of candidate variables for subsequent regression model testing. The rule for retaining variables was not simply p < 0.05. Rather, the decision whether to retain a variable was based on a combination of factors, including strength of association, prior research findings, and clinical importance as determined by investigators. Below, we outline the subsequent steps taken to optimize its clinical utility. Full descriptions of each step follow the outline.

1. We used a least absolute shrinkage operator (LASSO) logistic regression model for variable selection [42].

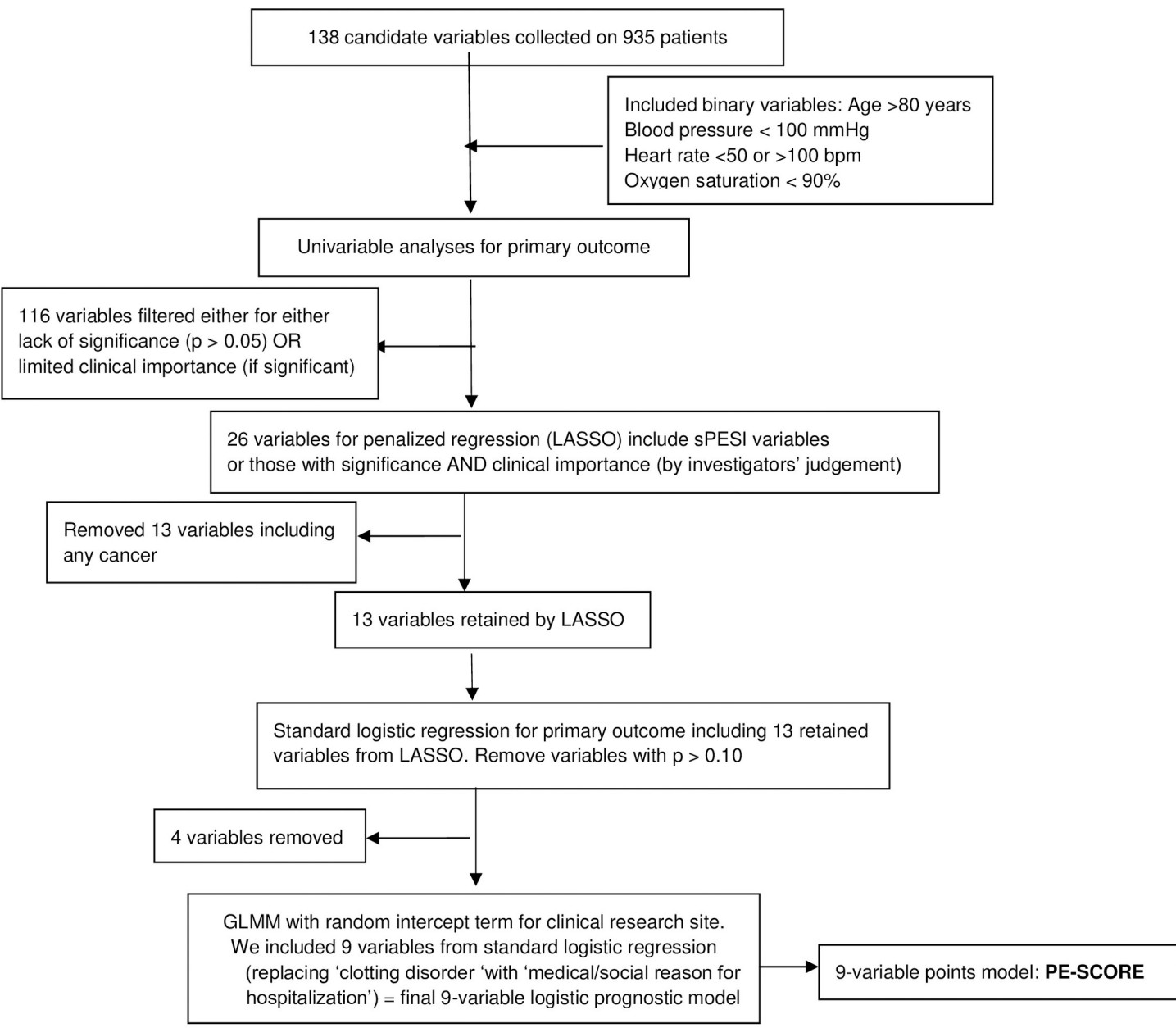

GLMM= generalized linear mixed model, sPESI = simplified pulmonary embolism severity index

**Fig 1. Deriving the nine-variable prognostic model.**

2. To further assess the predictor variables selected by the LASSO procedure, we included them in a standard logistic regression with the primary outcome as the response on the development data. We excluded predictor variables with $p > 0.10$ from further analysis.

3. We ran a generalized linear mixed model (GLMM) on the development database, with the primary outcome as the response and the reduced set of predictor variables identified by the LASSO and standard logistic regression models. The GLMM included a random intercept term for the clinical site to adjust for intra-site clustering.

4. To facilitate real-time clinical use by providers, we simplified the final 9-variable logistic regression model to a 9-variable points model that we named PE-SCORE.

Per the outline, we first used a LASSO logistic regression model for variable selection. LASSO is a type of penalized regression method that minimizes collinearity and avoids overfitting the model [43]. In addition, LASSO partitioned the development database such that two-thirds (67%) of data were used to train (or fit) the model, while 33% of the data were used for the first stage of internal validation of the model [44]. We selected the optimal level of penalization by using average squared error between responses and predictions in the internal validation data [44].

To further assess the predictor variables selected by the LASSO procedure, we included them in a standard logistic regression with the primary outcome as the response on the development data. We excluded predictor variables with p > 0.10 to create a more parsimonious model. Because of possible intra-site clustering, we considered the clinical research site to have a potentially important random effect in modeling the primary outcome. To assess site differences on the primary outcome and selected predictor variables, we used one-way analysis of variance for continuous variables, the Kruskal-Wallis test for ordinal variables, and the chi-square test for categorical variables. Informed by these findings, we ran a generalized linear mixed model (GLMM) on the development database, with the primary outcome as the response and the reduced set of predictor variables identified by the LASSO and standard logistic regression models. The GLMM included a random intercept term for the clinical research site to adjust for intra-site clustering. To determine the importance of the site effect in the model, we assessed its variance using a test based on the ratio of residual pseudo-likelihoods. We tested odds ratios of retained variables and used their confidence intervals (CIs) to determine significance as predictors of the primary composite outcome.

**Presentation of prediction model.** For the logistic regression, we reported coefficients for the variables in the final model, p-values, likelihood ratios, and odds ratios with confidence intervals. [The logistic regression equation is available for calculation of the probabilities.] Next, we assigned whole points and weights to the final variables of the tool, which were proportional to each variable's odds ratio for the primary outcome. We developed the points tool, called **P**ulmonary **E**mbolism **S**hort-term **C**linical **O**utcome **R**isk **E**stimator (PE-SCORE), for real-world usefulness to providers at the point of decision-making [2, 30, 45–47].

**External validation.** We used the external validation database to test the PE-SCORE model for reproducibility of results and to measure performance of the model on an entirely different sample. Site investigators and data extractors were blinded to the selection of development and validation databases.

We reported descriptive statistics to determine similarities and differences between the databases and compared them with t-test and chi-square analyses for predictor variables. We ran the points model on the validation database.

**Prognostic model performance.** We reported on the prognostic performance of both the logistic model and the points model (PE-SCORE) on the development and validation databases. We measured and assessed sensitivity, specificity, and positive and negative predictive values for the primary outcome (yes/no) using two thresholds (low-risk and high-risk) for the PE-SCORE model. To report on discrimination, we reported sensitivity, 1 minus specificity,

and receiver operating characteristic (ROC) to derive the area under the curve (AUC) and area under precision recall curve (AUCpr), with 95% confidence intervals and F1 scores and curves for visualization. For calibration, we 1) reported the proportion of observed actual events versus predicted probabilities, and 2) assessed goodness of fit between individuals with and without the outcome of interest with the Spiegelhalter z test and its p-value [48]. We reported measurements of calibration slope for overestimation and underestimation of risk prediction and the intercept for calibration-in-the- large [49, 50]. We provided figures of calibration curves for visualization [50, 51]. We used the following interpretation guideline: A slope < 1.0 suggests estimated risks are exaggerated, whereas slope > 1 suggests risks are underestimated. The calibration intercept was used for overall calibration-in-the-large. Using an optimal value of 0, negative values indicated overestimation, whereas positive values suggested underestimation.

To compare model performance, we compared the AUC of the full logistic model with the PE-SCORE in the development dataset. For this comparison, we used the method described by DeLong [52]. To compare AUC of PE-SCORE in the development and validation databases, we used the chi square test presented by Gonen [53].

## Results

### Participants

We enrolled 1008 patients into the development database, with 73 post-enrollment exclusions, leaving 935 records for analysis. We enrolled 815 patients in the validation database, with 14 post-enrollment exclusions, leaving 801 records for analysis. As shown in Table 1, patient characteristics in both databases were similar, as was the incidence of primary composite outcome and each of its components. Recurrence of VTE, major bleeding, and death within 30 days were higher in the development database.

There was low missingness for candidate variables. The variable with the most frequent missing responses (marked as absent) was GDE score at 2.2% and 3.4% in the development and validation databases, respectively. GDE missingness, however, was expected. Our

**Table 1. Descriptive statistics of development and validation databases.**

| | Development | Validation |
|---|---|---|
| | (N = 935) | (N = 801) |
| **Age** | | |
| Mean (SD) | 60.3 (16.5) | 58.5 (16.7) |
| Median [Min, Max] | 62.0 [18.0, 104] | 60.0 [19.0, 101] |
| **Age > 80** | 92 (9.8%) | 69 (8.6%) |
| **Initial Systolic Blood Pressure** | | |
| Mean (SD) | 132 (24.8) | 132 (23.7) |
| Median [Min, Max] | 133 [55.0, 223] | 131 [60.0, 210] |
| Missing | 0 (0%) | 4 (0.5%) |
| **Systolic Blood Pressure < 100 mmHg** | 82 (8.8%) | 54 (6.8%) |
| **Initial Heart Rate (beats/min)** | | |
| Mean (SD) | 98.8 (21.5) | 97.0 (21.3) |
| Median [Min, Max] | 98.0 [35.0, 184] | 96.0 [45.0, 182] |
| Missing | 1 (0.1%) | 4 (0.5%) |
| **Abnormal Heart Rate (< 50 or > 100 beats/min)** | 435 (46.6%) | 335 (42.0%) |
| **Shock Index Calculation** | | |
| Mean (SD) | 0.776 (0.248) | 0.764 (0.249) |
| Median [Min, Max] | 0.700 [0.300, 2.00] | 0.700 [0.300, 2.50] |
| Missing | 1 (0.1%) | 4 (0.5%) |

(*Continued*)

**Table 1.** (Continued)

| | Development | Validation |
|---|---|---|
| | (N = 935) | (N = 801) |
| **Initial Respiratory Rate** | | |
| Mean (SD) | 20.0 (4.66) | 19.9 (4.55) |
| Median [Min, Max] | 18.0 [10.0, 48.0] | 18.0 [11.0, 47.0] |
| Missing | 4 (0.4%) | 4 (0.5%) |
| **Initial Pulse Oximetry (%)** | | |
| Mean (SD) | 95.4 (4.88) | 95.6 (4.26) |
| Median [Min, Max] | 96.0 [37.0, 100] | 96.0 [67.0, 100] |
| Missing | 3 (0.3%) | 4 (0.5%) |
| **Initial Temperature (F)** | | |
| Mean (SD) | 98.1 (0.973) | 98.3 (0.886) |
| Median [Min, Max] | 98.1 [89.0, 103] | 98.2 [93.8, 103] |
| Missing | 20 (2.1%) | 7 (0.9%) |
| **Body Mass Index (kg/m$^2$)** | | |
| Mean (SD) | 31.2 (8.68) | 31.4 (9.22) |
| Median [Min, Max] | 29.8 [14.1, 79.9] | 30.0 [14.1, 87.6] |
| Missing | 18 (1.9%) | 59 (7.4%) |
| **Point-of-care** | | |
| **BNP Level, pg/ml** | | |
| Mean (SD) | 245 (487) | 303 (704) |
| Median [Min, Max] | 71.0 [4.00, 4670] | 76.0 [7.00, 8280] |
| Missing | 334 (35.7%) | 256 (32.0%) |
| **NT Pro BNP** | | |
| Mean (SD) | 1710 (5960) | 1030 (3000) |
| Median [Min, Max] | 159 [5.00, 70000] | 109 [6.00, 32500] |
| Missing | 644 (68.9%) | 577 (72.0%) |
| **Point-of-care** | | |
| **Troponin ng/L** | | |
| Mean (SD) | 0.860 (21.8) | 0.0928 (0.289) |
| Median [Min, Max] | 0.0300 [0, 648] | 0.0200 [0, 4.19] |
| Missing | 52 (5.6%) | 10 (1.2%) |
| **Length of Stay, days** | | |
| Mean (SD) | 4.7 (6.2) | 4.8 (5.0) |
| Missing | 40 (4.3%) | 27 (3.4%) |
| **Hospital Length of Stay less than 24 hours (discharged)** | 137 (14.6%) | 91 (11.3%) |
| **Clinical Research Site** | | |
| Carolinas Medical Center | 312 (33.4%) | 409 (51.1%) |
| San Diego | 189 (20.2%) | 66 (8.2%) |
| Vanderbilt University Medical Center | 134 (14.3%) | 78 (9.7%) |
| University of Utah | 78 (8.3%) | 85 (10.6%) |
| Orlando Regional Medical Center | 105 (11.2%) | 89 (11.1%) |
| Christiana Care | 117 (12.5%) | 74 (9.2%) |
| **Gender** | | |
| Female | 455 (48.7%) | 392 (48.9%) |
| Male | 480 (51.3%) | 409 (51.1%) |
| **Race** | | |
| Black | 253 (27.1%) | 255 (31.8%) |
| White | 638 (68.2%) | 497 (62.0%) |
| American Indian/Alaskan Native | 7 (0.7%) | 6 (0.7%) |
| Asian | 10 (1.1%) | 7 (0.9%) |
| Pacific Islander/Native Hawaiian | 2 (0.2%) | 1 (0.1%) |
| Unknown/Other | 25 (2.7%) | 35 (4.4%) |
| **Ethnicity** | | |
| Hispanic or Latino | 73 (7.8%) | 45 (5.6%) |
| Not Hispanic or Latino | 825 (88.2%) | 744 (92.9%) |
| Unknown | 36 (3.9%) | 12 (1.5%) |

(*Continued*)

**Table 1.** (Continued)

| | Development | Validation |
|---|---|---|
| | (N = 935) | (N = 801) |
| Missing | 1 (0.1%) | 0 (0%) |
| **Preceding Episode Syncope** | | |
| Yes | 92 (9.8%) | 68 (8.5%) |
| **Transient Hypotension** | | |
| Yes | 68 (7.3%) | 72 (9.0%) |
| **Preceding Bradycardia** | | |
| Yes | 16 (1.7%) | 12 (1.5%) |
| **Preceding Pulselessness** | | |
| Yes | 12 (1.3%) | 13 (1.6%) |
| **Prior diagnosis of PE or DVT** | | |
| Yes | 230 (24.6%) | 205 (25.6%) |
| **Family History of VTE** | | |
| Yes | 59 (6.3%) | 57 (7.1%) |
| **Hormone Replacement Therapy** | | |
| Yes | 28 (3.0%) | 29 (3.6%) |
| Missing | 1 (0.1%) | 0 (0%) |
| **Recent Pregnancy** | | |
| Yes | 8 (0.9%) | 12 (1.5%) |
| Missing | 2 (0.2%) | 0 (0%) |
| **Creatinine > 2.0 mg/dL** | | |
| Yes | 81 (8.7%) | 44 (5.5%) |
| Missing | 0 (0%) | 1 (0.1%) |
| **Moderate or severe liver disease** | | |
| Yes | 20 (2.1%) | 20 (2.5%) |
| **Clotting Disorders** | | |
| Yes | 27 (2.9%) | 24 (3.0%) |
| Missing | 0 (0%) | 5 (0.6%) |
| **Recent Hospitalization** | | |
| Yes | 286 (30.6%) | 300 (37.5%) |
| **Recent Trauma** | | |
| Yes | 66 (7.1%) | 60 (7.5%) |
| Missing | 0 (0%) | 2 (0.2%) |
| **Indwelling Vascular Catheter** | | |
| Yes | 56 (6.0%) | 60 (7.5%) |
| Missing | 0 (0%) | 3 (0.4%) |
| **Chronic Obstructive Pulmonary Disease** | | |
| Yes | 136 (14.5%) | 121 (15.1%) |
| **Any cancer** | | |
| Yes | 230 (24.6%) | 200 (25.0%) |
| **Heart Failure** | | |
| Yes | 55 (5.9%) | 71 (8.9%) |
| Missing | 0 (0%) | 1 (0.1%) |
| **Hemiplegia** | | |
| Yes | 22 (2.4%) | 26 (3.2%) |
| **Diabetes without end organ damage** | | |
| Yes | 109 (11.7%) | 114 (14.2%) |
| Missing | 0 (0%) | 2 (0.2%) |
| **Diabetes with end organ damage** | | |
| Yes | 58 (6.2%) | 43 (5.4%) |
| **Suspected Hypovolemia** | | |
| Yes | 44 (4.7%) | 39 (4.9%) |
| **Total Charlson Index** | | |
| 0 | 397 (42.5%) | 326 (40.7%) |
| 1 | 159 (17.0%) | 140 (17.5%) |
| 2 | 132 (14.1%) | 119 (14.9%) |

(*Continued*)

**Table 1.** (Continued)

| | Development | Validation |
|---|---|---|
| | (N = 935) | (N = 801) |
| 3 | 71 (7.6%) | 55 (6.9%) |
| 4 | 23 (2.5%) | 26 (3.2%) |
| 5 | 26 (2.8%) | 19 (2.4%) |
| 6 | 57 (6.1%) | 63 (7.9%) |
| 7 | 31 (3.3%) | 17 (2.1%) |
| 8 | 20 (2.1%) | 13 (1.6%) |
| 9 | 10 (1.1%) | 8 (1.0%) |
| 10 | 4 (0.4%) | 12 (1.5%) |
| 11 | 2 (0.2%) | 2 (0.2%) |
| 12 | 2 (0.2%) | 1 (0.1%) |
| 13 | 1 (0.1%) | 0 (0%) |
| **Other Medical or Social reason for treatment in Hospital >24 hours** | | |
| Yes | 526 (56.3%) | 310 (38.7%) |
| Missing | 5 (0.5%) | 14 (1.7%) |
| **Natriuretic peptide elevation** | | |
| Yes | 349 (37.3%) | 308 (38.5%) |
| Missing | 42 (4.5%) | 31 (3.9%) |
| **Troponin Elevation** | | |
| Yes | 269 (28.8%) | 185 (23.1%) |
| Missing | 12 (1.3%) | 9 (1.1%) |
| **CT RV:LV Ratio** | | |
| Yes | 309 (33.0%) | 249 (31.1%) |
| Missing | 17 (1.8%) | 20 (2.5%) |
| **Most Proximal Location thrombus on CTPA** | | |
| Saddle | 106 (11.6%) | 92 (11.6%) |
| Proximal pulmonary artery | 192 (21.0%) | 102 (12.9%) |
| Lobar | 324 (35.5%) | 281 (35.6%) |
| Segmental | 245 (26.8%) | 256 (32.4%) |
| Subsegmental | 46 (5.0%) | 59 (7.5%) |
| **GDE Score** | | |
| 0 | 604 (64.6%) | 549 (68.5%) |
| 1 | 72 (7.7%) | 62 (7.7%) |
| 2 | 122 (13.0%) | 104 (13.0%) |
| 3 | 116 (12.4%) | 59 (7.4%) |
| Missing | 21 (2.2%) | 27 (3.4%) |
| **Poor LV Function** | | |
| Yes | 68 (7.3%) | 63 (7.9%) |
| Missing | 19 (2.0%) | 47 (5.9%) |
| **GDE Showing abnlRV** | | |
| Yes | 310 (33.2%) | 225 (28.1%) |
| Missing | 21 (2.2%) | 27 (3.4%) |
| **If GDE >0, is it acute?** | | |
| Yes | 278 (63.2%) | 191 (65.0%) |
| No | 121 (27.5%) | 69 (23.5%) |
| Indeterminate | 41 (9.3%) | 34 (11.6%) |
| **Low-risk sPESI** | | |
| Yes | 314 (33.6%) | 297 (37.1%) |
| **Low-risk ESC** | | |
| Yes | 77 (8.2%) | 106 (13.2%) |
| **Primary Composite Outcome** | | |
| Yes | 209 (22.4%) | 213 (26.7%) |
| **Secondary Outcome** | | |
| Yes | 331 (35.4%) | 313 (39.1%) |
| Missing | 5 (0.5%) | 0 (0%) |
| **Death within 5 days** | | |

(*Continued*)

**Table 1.** (Continued)

| | Development | Validation |
|---|---|---|
| | (N = 935) | (N = 801) |
| Yes | 24 (2.6%) | 16 (2.0%) |
| **Cardiac Arrest within 5 days** | | |
| Yes | 15 (1.6%) | 17 (2.1%) |
| **Respiratory Failure within 5 days** | | |
| Yes | 78 (8.3%) | 71 (8.9%) |
| **Dysrhythmia within 5 days** | | |
| Yes | 60 (6.4%) | 53 (6.6%) |
| **Major Bleeding within 5 days** | | |
| Yes | 23 (2.5%) | 26 (3.2%) |
| **Reperfusion intervention within 5 days** | | |
| Yes | 60 (6.4%) | 63 (7.9%) |
| **Hypotension Pressors within 5 days** | | |
| Yes | 46 (4.9%) | 35 (4.4%) |
| **Hypotension Fluid within 5 days** | | |
| Yes | 72 (7.7%) | 101 (12.6%) |
| **Hypoxia within 5 days** | | |
| Yes | 423 (45.2%) | 364 (45.4%) |
| **Recurrence of VTE within 30 days** | | |
| Yes | 12 (1.3%) | 10 (1.2%) |
| Missing | 13 (1.4%) | 1 (0.1%) |
| **Major Bleeding within 30 days** | | |
| Yes | 30 (3.2%) | 37 (4.6%) |
| Missing | 12 (1.3%) | 1 (0.1%) |
| **Death within 30 days** | | |
| Yes | 68 (7.3%) | 56 (7.0%) |
| Missing | 3 (0.3%) | 0 (0%) |
| **Active Bleeding** | | |
| Yes | 0 (0%) | 105 (13.1%) |
| Missing | 935 (100%) | 14 (1.7%) |
| **Anticoagulation Initiated** | | |
| Yes | 868 (92.8%) | 714 (89.1%) |
| Missing | 1 (100%) | 12 (1.5%) |

Abbreviations: BNP = brain natriuretic peptide; PE = pulmonary embolism; DVT = deep vein thrombosis; VTE = venous thromboembolism; ESC = European Society of Cardiology Pulmonary Embolism Management guidelines (2019)[15]; CT = computed tomography; LV = left ventricle; RV = right ventricle; GDE = goal-directed echocardiography; sPESI = simplified Pulmonary Embolism Severity Index

assessment showed the impact of missing GDE values on outcomes was minimal: the percentage of patients experiencing the primary outcome for those with GDE negative, positive, and missing responses for abnlRV were 14.4%, 38.4%, and 28.6%, respectively (S1 Table). Twenty-one (2.2%) patients in the developmental database were missing GDE. Most of these missing GDE scores were marked as inadequate for interpretation. Six of the 21 had a positive primary outcome. For the combined databases, 1706 GDE were performed by faculty (29.1%), fellows (9.7%), third-year emergency medicine (EM) residents (23.6%), second-year EM residents (29.1%), and first-year EM residents (15.1%). There were 48 patients (2.8%) without GDE scores: 18 (37.5%) were performed but technically difficult and not interpretable; whereas GDE was not performed on 30 patients (62.5%) before ED discharge.

Enrollments were not evenly distributed for the six sites in both databases. The central site enrolled 33.4% of the development database and 51.1% of the validation database. The other sites enrolled 8.3%–20.2% and 8.2%–11.1% in the two databases, respectively.

## Model development

S1 Table shows main results of univariable analyses of candidate variables on the development database. Notably, any cancer (p = 0.987) and heart failure (p = 0.285) had non-significant p-values. Twenty-six of the 138 candidate variables vetted by univariable analyses had p-values below 0.05 and were retained for subsequent LASSO regression. We re-entered variables for chronic obstructive pulmonary disease (COPD), cancer, and oxygen saturation below 90% because these were variables in validated sPESI and Hestia models. LASSO retained 13 variables; cancer was not retained again. We next ran a standard logistic regression with the 13 retained variables, nine of which had p < 0.10 in the logistic model and were retained for further analysis.

In the univariable comparisons of clinical research sites, we found statistically significant differences between sites for the variables shown in S2 Table (primary composite outcome, race, age, ethnicity, abnormal heart rate, creatinine greater than 2.0 mg/dL, abnormal RV by imaging, and medical/social reasons for hospitalization). Moreover, the random intercept term for the clinical research site was statistically significant (p < 0.01) in the GLMM. Accordingly, we retained 'clinical research site' as a random effect in the model. [Although clotting disorder was statistically significant, it was uncommon; thus, it was not included in the final prognostic model.] Table 2 shows the nine variables used in the final logistic regression equation.

## Model specification

The logistic regression equation to determine probability of the primary outcome is $P = [1 + \exp(-(\alpha_{RE} + \Sigma_i \beta_i x_i))]^{-1}$, where $\alpha RE$ is the fixed intercept (-2.91) and $\Sigma_i \beta_i x_i$ is the sumproduct of the nine fixed regression coefficients of the random effects model.

**To convert the 9-variable logistic regression prognostic model into a simpler format for usefulness, we used the odds ratios shown in Table 3.** The odds ratios of most of the nine predictor variables were similar and each was assigned 1 point, except for the creatinine > 2.0 mg/dL variable, which was assigned 2 points. The reason 2 points were awarded for creatinine > 2.0 mg/dL was based on the adjusted odds ratio of 5.37 for this variable. The adjusted odds ratio of 5.37 for renal impairment was more than double that of 5 variables in the model. Compared to dysrhythmia, which had the second highest adjusted odds of 4.00, the

**Table 2. Final variables of logistic regression model.**

| Predictor | Adjusted Odds | 95% CI Odds Ratio | | Coefficient | 95% CI Coefficient | |
|---|---|---|---|---|---|---|
| | | Lower | Upper | | Lower | Upper |
| Creatinine > 2.0 mg/dL | 5.37 | 2.49 | 11.58 | 1.68 | .911 | 2.45 |
| Dysrhythmia | 4.00 | 2.07 | 7.73 | 1.39 | .730 | 2.04 |
| Suspected/confirmed systemic infection | 3.47 | 1.64 | 7.37 | 1.24 | .491 | 2.00 |
| Systolic Blood Pressure < 100 | 2.87 | 1.63 | 5.07 | 1.05 | .486 | 1.62 |
| Abnormal Heart rate | 2.26 | 1.52 | 3.35 | .813 | .418 | 1.21 |
| Preceding episode of syncope | 1.97 | 1.15 | 3.38 | .680 | .141 | 1.22 |
| Medical social reason for hospitalization | 1.91 | 1.21 | 3.03 | .649 | .190 | 1.11 |
| Echocardiography RV abnormal | 1.81 | 1.12 | 2.91 | .592 | .115 | 1.07 |
| CT RV:LV ratio elevated | 1.73 | 1.05 | 2.84 | .548 | .050 | 1.05 |
| Intercept | | | | -2.91 | -4.01 | -1.80 |

**Abbreviations:** CI = confidence interval; RV = right ventricle; LV = left ventricle; CT = computed tomography

**Table 3. Primary outcome probability for final model variables.**

| Final Predictor Variable | Adjusted Odds Ratio | Development Database Relative Risk | Validation Database Relative Risk | Points Assigned |
|---|---|---|---|---|
| Creatinine > 2.0 mg/dL | 5.37 | 2.48 | 2.16 | 2 |
| Dysrhythmia | 4.00 | 2.39 | 3.67 | 1 |
| Suspected/confirmed systemic infection | 3.47 | 2.63 | 3.67 | 1 |
| Systolic blood pressure < 100 mmHg | 2.87 | 2.65 | 2.85 | 1 |
| Abnormal heart rate (<50 or >100 beats/min) | 2.26 | 2.17 | 1.67 | 1 |
| Syncope | 1.97 | 2.00 | 2.25 | 1 |
| Medical or social reason for hospitalization | 1.91 | 2.00 | 1.76 | 1 |
| Echocardiography with abnormal RV | 1.81 | 2.67 | 3.16 | 1 |
| CT RV:LV ratio elevated | 1.73 | 2.23 | 2.38 | 1 |
| Total Points | | | | |
| PE-SCORE score (minimum = 0; maximum = 10 points) | | | | |

**Abbreviations:** CT = computed tomography; LV = left ventricle; RV = right ventricle

adjusted odds for creatinine elevation was 40% higher. We recognized that by awarding 2 points for creatinine elevation, the range for our point system would be 0–10, which is standard for many similar scales. The weights assigned to each variable in the final PE-SCORE model are listed in Table 3. The lowest PE-SCORE is 0 and the highest score is 10.

## Presentation of points prognostic model

S1 Fig illustrates a useful presentation and application platform of the points model. With PE-SCORE, a provider can list the 9 variables and whether the findings for each are present or not (yes or no). If creatinine greater than 2.0 mg/dL is present, 2 points are awarded. For the other 8 variables, 1 point is awarded if the condition is present. If any provider wants to use a finer scale, we have supplied the coefficients derived from the logistic regression model. With the logistic regression equation, a computer program would be required to calculate the probability of a positive primary outcome.

## External validation

Table 4 shows the actual versus predicted events of PE-SCORE on the validation database. Predicted events were derived from the logistic regression model estimations. At the low end of the risk estimation, actual events in the validation database were higher (8% compared to 2% in the development database). There were no deaths within 5 days for patients with PE-SCORE of zero. There was one death among patients with PE-SCORE of 4, but it was not considered PE-related (segmental PE with coexisting perforated intestinal ulcer and gastrointestinal bleeding). The patient did not have CT or GDE finding of RV abnormalities, although both troponin and BNP were elevated. In this case, the PE-SCORE was elevated (although the sPESI was zero) because of other medical conditions, a heart rate of 105 bpm, and creatinine greater than 2.0 mg/dL.

## Prognostic model performance

All 9 components of the prognostic model were available for full scoring of PE-SCORE for 888 of 935 patients (95%) in the development database and 737 of 801 patients (92%) in the validation database. In the development database, for the minimum score of zero, the proportion

**Table 4. Number of predicted and actual events in validation database.**

| PE-SCORE | Predicted from frequency in developmental database | | Predicted Events | Actual Events |
|---|---|---|---|---|
| | % Positive Primary Outcome | % Positive Primary Outcome | | |
| | Development | Validation | | |
| 0 points | 2.05 | 8.11 | 3.79 | 15 |
| 1 point | 7.31 | 16.72 | 13.89 | 31 |
| 2 points | 18.59 | 23.72 | 29.00 | 37 |
| 3 points | 38.00 | 42.43 | 39.52 | 44 |
| 4 points | 35.58 | 58.57 | 24.21 | 41 |
| 5 points | 63.83 | 85.71 | 13.40 | 18 |
| 6 + points | 69.60 | 100 | 7.66 | 11 |

with primary composite outcome was 2%. Among those with scores of 6 or higher, the composite outcome was 69.6%. The exception was 38% for a score of 3 and 35.6% for a score of 4. In the validation dataset, for the minimum score of 0, the proportion with primary outcome was 8%. Among those with scores of 6 or higher, 100% had the primary outcome. The discrepancy in the middle ranges was absent. Based on the results, we set a low-risk threshold for PE-SCORE at 0 points and high-risk threshold at 5 points.

Table 5 shows performance of a) the logistic regression model on the development database, and b) the PE-SCORE model on the development and validation databases (AUC 0.78 and 0.77, respectively). Fig 2 shows AUC for the logistic regression model on the development database, followed by AUC for PE-SCORE on the development and validation databases. The AUC of the full logistic model (AUC 0.83) and PE-SCORE (AUC 0.78) were compared in the development dataset with DeLong's test with p-value <0.01, indicating a significant difference in the two ROC curves. Although the AUC of PE-SCORE was less than that of the logistic regression model, the prognostic performance of both logistic regression and PE-SCORE are in the good range. Next, the chi square test comparison of ROC curves of PE-SCORE in the development dataset (AUC 0.78) versus validation dataset (AUC 0.77) resulted in a p-value of 0.49, indicating no statistically significant difference in the two ROC curves. The prognostic performance of PE-SCORE was similar in the development and validation databases.

We report on the AUCpr due to the imbalance in outcomes on both development and validation databases [54]. Fig 3 shows AUCpr for logistic regression model on the development database, followed by the AUCpr for PE-SCORE on the development and validation databases.

**Table 5. Discrimination and calibration metrics.**

| Model | Discrimination | | Calibration | | | |
|---|---|---|---|---|---|---|
| | Sensitivity vs 1-specificity plot | Precision Recall curve | Spiegelhalter z test and p-value | Slope (95% CI) | Intercept (95% CI) | Hosmer-Lemeshow |
| | AUC (95% CI) | AUCpr (95% CI) | | | | p value |
| Logistic regression (development database) | 0.83 (0.80, 0.86) | 0.61 (0.57, 0.64) | 0.2933, 0.7693 | 1.029 (0.920, 1.138) | -0.006 (-0.040, 0.027) | 0.08 |
| PE-SCORE (development database) | 0.78 (0.75, 0.82) | 0.50 (0.39, 0.60) | -0.071, 0.9431 | 0.966 (0.829, 1.102) | 0.008 (-0.031, 0.047) | 0.01 |
| PE-SCORE (validation database) | 0.77 (0.73, 0.81) | 0.63 (0.43, 0.81) | 0.3283, 0.7427 | 1.006 (0.867, 1.146) | -0.002 (-0.049, 0.045) | 0.76 |

Abbreviations: AUC = area under the curve, CI = confidence interval

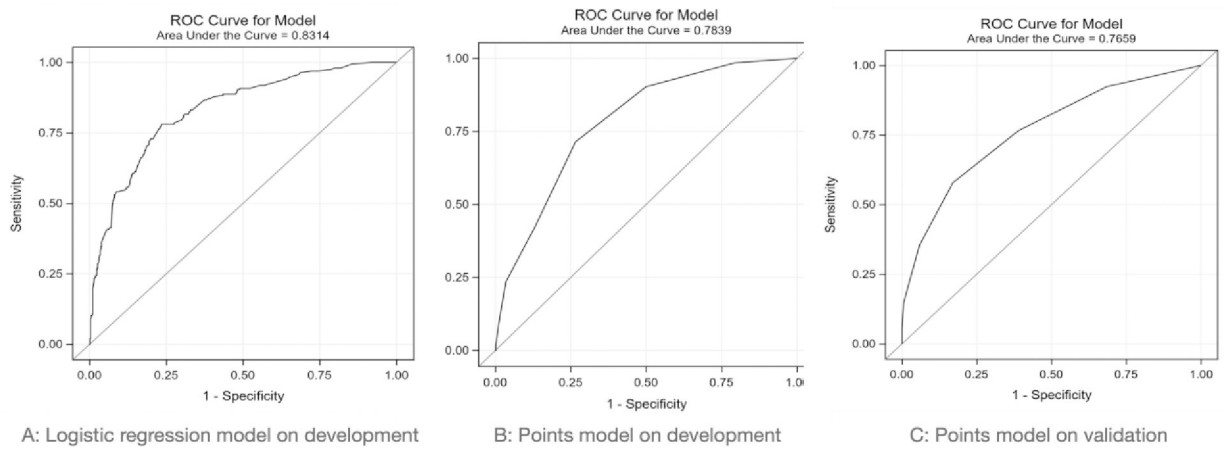

**Fig 2.** Area under receiver operating characteristic of: A) logistic regression model on the development database; B) PE-SCORE model on development database; and C) PE-SCORE model on validation database.

In Table 5, we provide four metrics for calibration of PE-SCORE on the development and validation dataset and for logistic regression model on the development database. Fig 4 shows calibration slope and intercept values to be excellent: 1) the Spiegelhalter z test did not indicate lack of fit (p > 0.05); 2) calibration curve slope values were close to 1.0 and linear regression intercept values were close to zero. Calibration slopes and intercepts were close to 1.0 and zero on both databases. Although the Hosmer-Lemeshow test suggested lack of fit (p <0.1) for the full regression model and points model on the development database, those results were offset by three calibration test metrics that indicated excellent calibration [50]. Fig 5 shows the proportion experiencing the primary composite outcome (death or clinical deterioration event) at each total PE-SCORE and the number of patients experiencing death for the primary composite in both databases.

Prognostic values [positive predictive value (PPV) and negative predictive value (NPV)] are affected by prevalence of the outcome of interest. Of 935 patients in the development database,

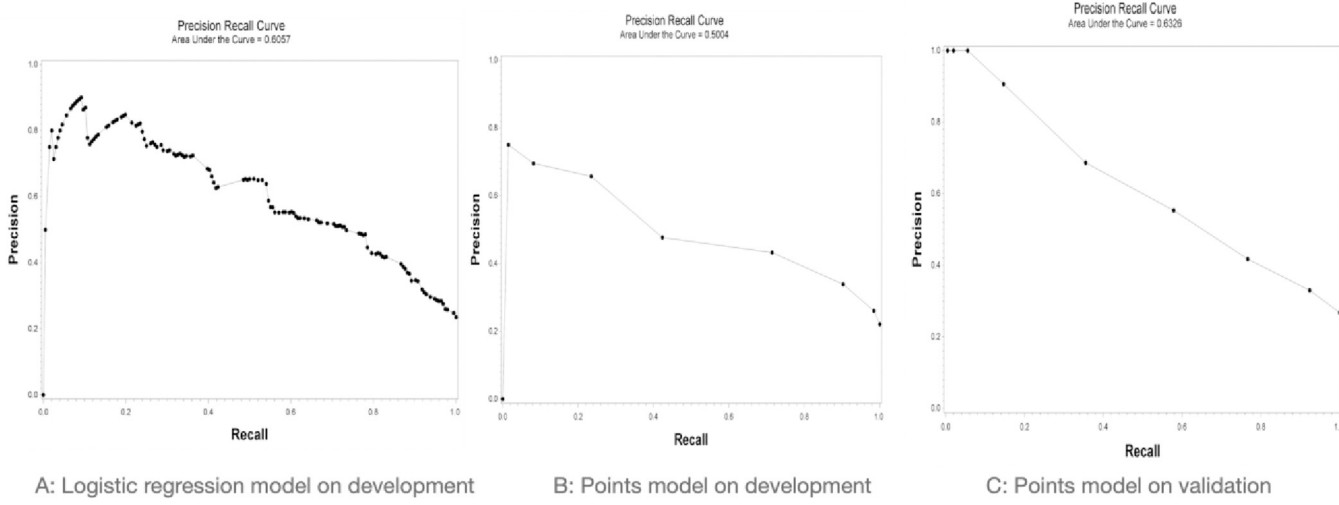

**Fig 3.** Precision recall curves of: A) logistic regression model on the development database; B) PE-SCORE model on the development database; and C) PE-SCORE model on the validation database.

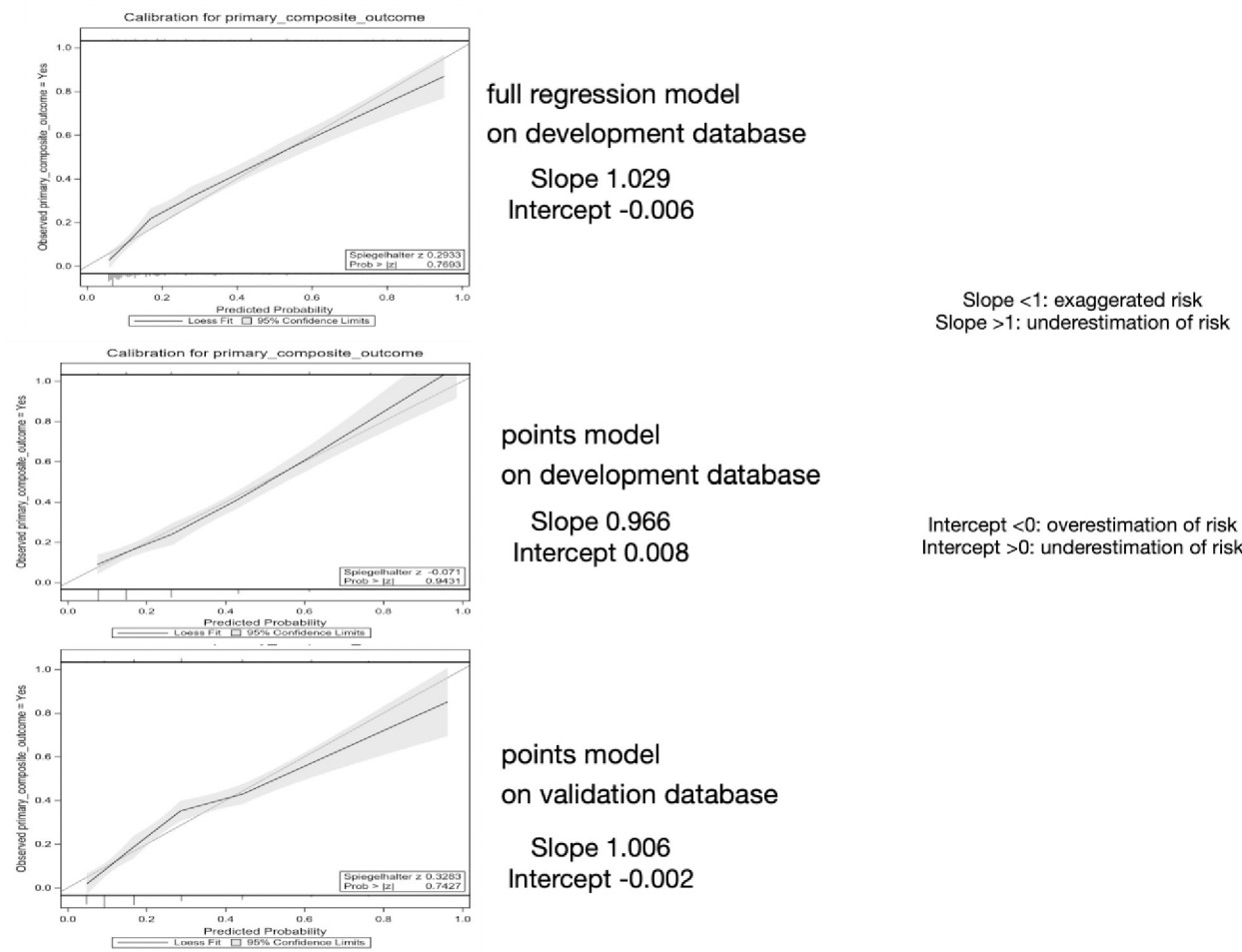

**Fig 4. Calibration curves of logistic regression on development and PE-SCORE model on both databases.**

888 (95%) had complete responses for nine components of the PE-SCORE tool. Of 801 patients in the validation database, 737 (91%) had complete responses for all nine components of the PE-SCORE tool. GDE was the only variable missing. None of the other 8 variables used to calculate the PE-SCORE were missing. A modified PE-SCORE that did not include GDE was calculated with a reduced potential range of 0–9 points. Their modified scores had an actual range of 0–4. The percentages of patients experiencing the primary outcome among those with modified PE-SCORE scores of 0, 1, 2, 3, 4 were 16.7%, 16.7%, 50.0%, 50.0%, and 0%, respectively. In comparison, the percentages of patients without missing GDE who experienced the primary outcome were 2.1%, 7.3%, 18.6%, 38.0%, and 35.6% among those with a PE-SCORE of 0, 1, 2, 3 and 4, respectively. Except for a modified score of 4, these percentages were higher in each point category for the patients with GDE missing than the same score for the group not missing GDE.

Table 6 shows traditional prognostic accuracy performance metrics for PE-SCORE (at two different risk thresholds) on the development and validation databases. We used a threshold of zero points for PE-SCORE to address low-risk stratification. A threshold of 5 for PE-SCORE indicates high-risk for clinical deterioration. At the lower-risk threshold, providers are most interested in the negative predictive value (NPV) of a prognostic model. We report on the model's performance in low- versus high-risk stratification because the decisions made are

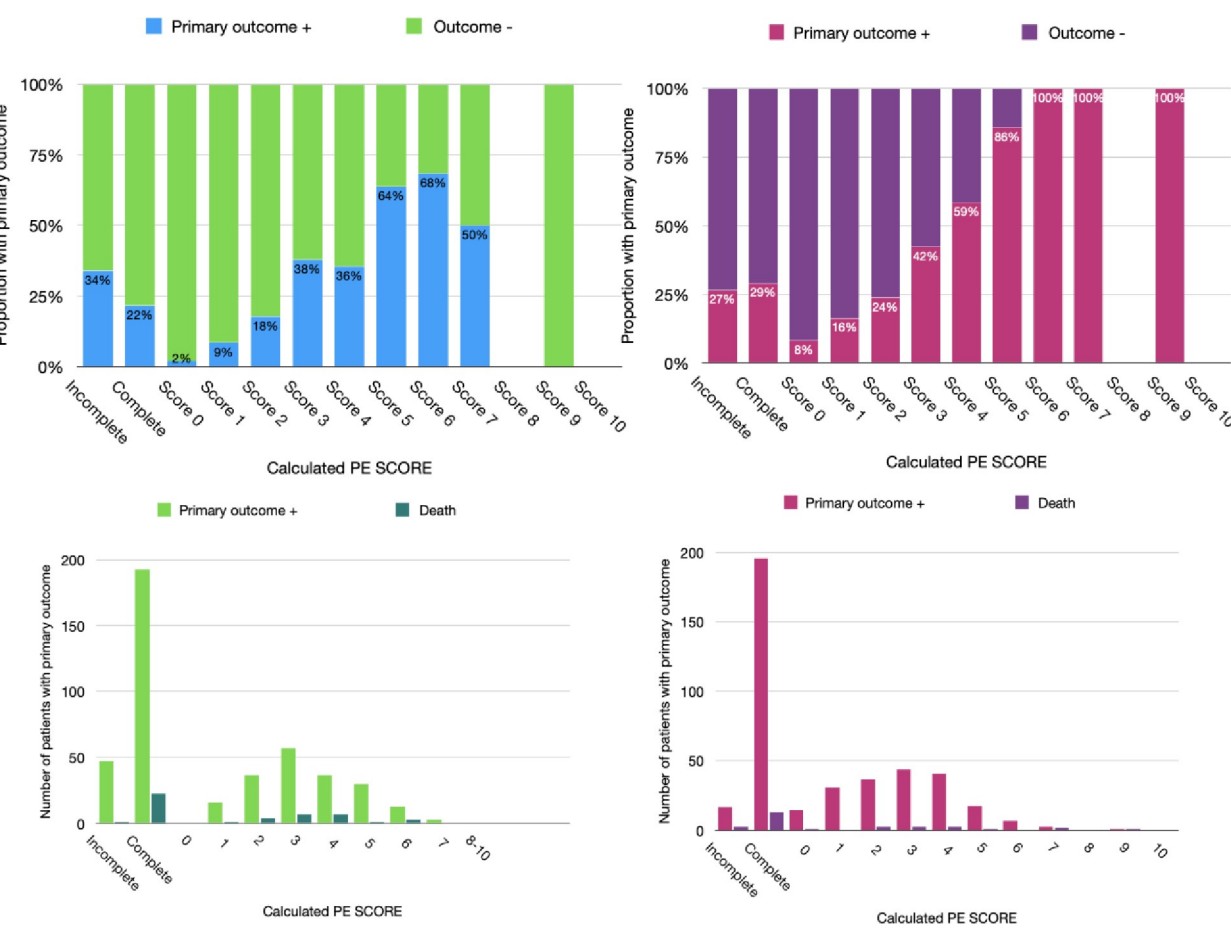

**Fig 5. Proportions with primary outcome by calculated PE-SCORE.** Legend: Panels A and B show 2D stacked column charts stratified by the proportions of patients with primary composite outcome positive (lower column) and the outcome negative groups (upper column) for each PE-SCORE calculation in the development and validation databases. Panels C and D show column charts for the number of patients with primary outcome positive next to the number with death for each PE-SCORE calculation in the development and validation databases.

quite different. Low-risk stratification increases consideration for immediate outpatient clinical management, whereas high-risk stratification increases the intensity of monitoring.

Low-risk PE-SCORE had sensitivity 98.5% (95.2%–99.6%) and 92.4% (87.5%–95.5%), specificity 20.7% (17.7%–23.9%) and 31.5% (27.6%–35.6%), PPV 26.0% (22.9%–29.3%) and 33.0% (29.1%–37.1%), and NPV 97.9% (93.6%–99.5%) and 91.9% (86.7%–95.2%), respectively. In addition, precision, recall, and F1 metrics are presented at both risk thresholds.

S3 Table shows the performance of sPESI and the 2019 version of the European Society of Cardiology (ESC) PE management guidelines at low-risk threshold on the two databases [15]. The sPESI represents a prognostic model without RV assessment variables and ESC represents an updated risk stratification model that combines a previously validated clinical prediction model (sPESI) with imaging RV assessment variables (using our definitions). We used it for the primary composite outcome of death and pre-defined clinical deterioration outcomes within 5 days. We identified low-risk ESC criteria, which incorporated low-risk sPESI threshold and absence of RV abnormalities (using our definitions). In the development and validation databases, low-risk sPESI had sensitivity 85.2% (79.6%–89.7%) and 80.4% (74.4%–85.5%), specificity 39.0% (35.4%–42.6%) and 43.4% (39.4%–47.6%), PPV 28.7% (27.0%–30.4%) and 34.1% (32.0%–36.3%), NPV 90.1% (86.7%–92.8%) and 85.9% (82.0%–89.0%) respectively. In

**Table 6. Performance of PE-SCORE model at two risk thresholds on both databases.**

| Low-risk threshold PE-SCORE cut-off = 0 points | | |
|---|---|---|
| *Development database* | Accuracy = 37.8% (34.6%–41.1%) | |
| Score+ (1–9 points) | sensitivity 98.5% (95.2%–99.6%) | PPV 26.0% (22.9%–29.3%) |
| Score- (0 points) | specificity 20.7% (17.7%–23.9%) | NPV 97.9% (93.6%–99.5%) |
| A = 193 | Precision = A/(A+C) = 0.26 | |
| | Recall = A/A+B) = 0.9847 | |
| B = 3 | F1 = (precision*Recall)/ (Precision + Recall) = 0.2056 | |
| C = 549 | | |
| D = 143 | | |
| *Validation database* | Accuracy = 47.8% (44.1%–51.4%) | |
| Score+ (1–9 points) | sensitivity 92.4% (87.5%–95.5%) | PPV 33.0% (29.1%–37.1%) |
| Score—(0 points) | specificity 31.5% (27.6%–35.6%) | NPV 91.9% (86.7%–95.2%) |
| A = 182 | Precision = A/(A+C) = 0.33 | |
| B = 15 | Recall = A/A+B) = 0.92 | |
| C = 370 | F1 = (precision*Recall)/ (Precision + Recall) = 0.24 | |
| D = 170 | | |
| High-risk threshold PE-SCORE cut-off = 5+ points | | |
| *Development database* | Accuracy 80.4% (77.6%–83.0%) | |
| Score+ (5–9 points) | sensitivity 23.5% (17.9%–30.1%) | PPV 65.7% (53.3%–76.4%) |
| Score- (0–4 points) | specificity 96.5% (94.8%–97.7%) | NPV 81.7% (78.8%–84.2%) |
| A = 46 | Precision = A/(A+C) = 0.65 | |
| B = 147 | Recall = A/A+B) = 0.24 | |
| C = 24 | F1 = (precision*Recall)/ (Precision + Recall) = 0.17 | |
| D = 671 | | |
| *Validation database* | Accuracy 76.8% (73.6%–79.8%) | |
| Score+ (5–9 points) | sensitivity 14.7% (10.2%–20.6%) | PPV 90.6% (73.8%–97.5%) |
| Score- (0–4 points) | specificity 99.4% (98.3%–99.9%) | NPV 76.2% (72.8%–79.2%) |
| A = 29 | Precision = A/(A+C) = 0.91 | |
| B = 168 | Recall = A/A+B) = 0.15 | |
| C = 3 | F1 = (precision*Recall)/ (Precision + Recall) = 0.13 | |
| D = 537 | | |

*Abbreviations for number of predicted events versus actual events*: A = True positive, B = False positive (Type II error), C = False positive (type I error), D = True negative.
**Other abbreviations:** PPV = positive predictive value; NPV = negative predictive value.

the development and validation databases, low-risk ESC sensitivity 99.5% (97.4%–99.9%) and 97.7% (94.6%–99.2%), specificity 10.5% (8.3%–12.9%) and 17.2% (14.2%–20.5%), PPV 24.2% (23.8%–24.7%) and 30.1% (29.2%–31.0%), NPV 98.7% (91.4%–99.8%) and 95.3% (89.3%–98.0%), respectively. Overall, both ESC and PE-SCORE models, which involved RV assessments, outperformed low-risk SPESI when focused on the primary composite outcomes that matter to point-of-care decision makers.

## Discussion

We used prospective registry databases and developed and validated an original prognostic model from a field of 138 candidate variables. The registry involved contemporaneous and early assessments for PE provoked RV abnormalities with predefined laboratory and imaging assessments, and focused on outcomes of interest to providers at the point of decision-making

and to pulmonary embolism response teams [5, 7, 13, 28]. The final variables in the prognostic model are readily available during ED evaluation, including interview questions, witnessed events (syncope), vital signs at presentation or on a cardiac monitor (heart rate and systolic blood pressure), past medical history, routine laboratory findings, and imaging. The two imaging variables were CT RV:LV ratio, which is determined from CT images, and goal directed echocardiography, which is performed at the patient's bedside and provides multiple dynamic images of the RV.

In a meta-analysis of 71 prognostic model reports, 17 were original prognostic models like our study [4]. The other 54 reports were validating, updating, or investigating the impact of prognostic models. For the 17 original prognostic studies, the number of candidate variables ranged from seven to greater than 30. In five studies, the number of candidate variables were either unclear or not reported [16, 55–58]. Few studies included imaging findings as candidate variables: echocardiography finding of abnlRV (one study), RV:LV ratio (two studies), CT findings of RV abnormality (one study), and ultrasound for venous thrombosis (four studies) [1, 16, 57–61].

Most reports on prognostic models for acute PE focus on outcomes of death, recurrent VTE, and bleeding at a time point of 30 days or longer [4, 18, 62]. In contrast, our study focused on death or clinical deterioration within five days of PE diagnosis, as outcomes that are important to providers and researchers [5, 7, 13, 25].

Prognostic performance of the logistic regression and PE-SCORE models was strong for discrimination and calibration. The logistic regression model had an AUC of 0.83 in the development database. The user-friendly PE-SCORE points tool had AUCs of 0.77 and 0.78 in the development and validation databases, respectively. When decision-making priority is focused on patient candidacy for outpatient treatment, PE-SCORE set to a low-risk threshold has high negative predictive value. When the decision-making priority is determining whether increased intensity of monitoring or increased considerations for escalated PE treatment may be indicated, PE-SCORE set to a high-risk threshold has moderate accuracy.

Regression analyses provide plausible ranking of importance of RV imaging variables in PE risk stratification: GDE had greater odds ratio than CT. We used GDE instead of comprehensive echocardiography to visually detect PE-provoked abnormalities of RV size, pressure, and systolic function. To assign a GDE score of 1 or more, providers were required to detect RV dilatation (not severe RV systolic dysfunction or septal shift alone). The ordinal nature of GDE scoring was itself calibrated, showing increased odds of clinical deterioration as GDE scores increased.

The absence of variables in our final prognostic model deserves discussion. Troponin and natriuretic peptides are considered influential PE prognostic predictors in meta-analyses [18, 22, 63–65]. Although our study's univariable analyses showed significant differences in both cardiac biomarkers in outcome groups, neither troponin nor natriuretic peptide elevation were retained after regression analyses. Our findings rank the predictive accuracy of laboratory RV assessments lower than imaging RV assessments in a restricted prognostic model. It is plausible natriuretic peptide and troponin do not directly identify the cardiac chamber experiencing acute myocardial stretch and myocardial injury. In contrast, GDE directly identifies RV dilatation and abnlRV systolic function. Age and cancer (predictors featured in models like PESI, sPESI, Hestia, and ESC) were not significant in univariable analyses or with penalized regression analysis. In the original PESI study, those aged > 65 years accounted for 52%–59% of the development and validation cohorts [2]. In our study, the proportion of patients aged 65 or older in both databases was lower at 39.3%. In the original PESI derivation and validation report, cancer was present in 19% and 16% of the databases, respectively. In our

PE-SCORE study, cancer was present in 24.8% of PE patients, but did not reach statistical significance for prognosis of acute clinical deterioration.

The absence of advanced age or cancer as discrete variables in the PE-SCORE does not prevent these features from being considered by provider's discretion for social or medical reasons for hospitalization or an increased level of monitoring—an important component of PE-SCORE. Original clinical scores or guidelines, which were developed for outcomes of death, recurrent VTE, or major bleeding 30 days or later, tend to be pragmatically modified or adapted to consider other social/medical conditions or laboratory or imaging findings instead of being used in isolation during clinical practice [7, 13, 15, 66]. With PE-SCORE, variables for provider discretion on social or medical reasons for hospitalization for increased monitoring and RV imaging assessment are built in.

Unlike our findings, some studies found troponin and echocardiography findings of abnlRV did not have prognostic value in determining in-hospital adverse events [67]. Zondag et al. reported that although 35% classified as low-risk by Hestia criteria had coexisting RV abnormality by CTPA, there was no difference in outcomes compared to patients without abnlRV [68].

Our study has several limitations. Although the validation was performed on a different database with data collected during a different time period, external validation should be conducted at sites outside the current registry. Our study focused on clinical deterioration and early mortality due to PE *severity*. We did not assess outcomes due to PE *treatment* (e.g., bleeding, bleeding risk, compliance with treatment), which would influence disposition decisions and need for safety outcomes. The study setting was focused on emergency department patients and ambulatory care settings where the cadence and feasibility of testing may not be generalizable to patients developing acute PE while already in the inpatient setting. Already hospitalized patients who develop acute PE may have different risk factors or susceptibilities to PE-associated deterioration from those diagnosed in an outpatient setting.

Our a priori study design included using troponin measurements as continuous data; however, institutional change in troponin assay at the central site interrupted plans to perform linear regression on the troponin variable. Similarly, two of the six sites used NT proBNP, while others used point-of-care BNP assay measurements. Therefore, we used institutional assay cut-offs to create categorical variables (troponin and natriuretic peptide elevation). Univariable analyses showed significant differences in mean troponin, point-of-care BNP, and NT proBNP measurements between outcome groups in both databases. Valuable information, however, may have been lost by converting a continuous variable into a categorical variable [69].

Univariable analysis identified the clinical site itself as a variable of importance. The logistic regression model therefore has a random effects intercept for clinical sites. The random intercept cannot be used in a risk calculation on patients at sites outside of the six sites of this study, as the random effect of the new site is unknown. Thus, only the fixed intercept of the random effects model is used in the risk calculation. Model performance at a clinical site outside the six sites in this study may differ. Other discrete variables that may be of interest (e.g., median income, insurance status, other social determinants of health) were not included in this study. Despite significant differences in patient characteristics between sites, the prognostic model performed well on patients.

Another possible limitation of our report is that machine-based learning derivation techniques may offer better management of multiple variables (including those with interactions); however, our preliminary steps with classification tree analysis were not helpful. The logistic regression model we developed had an AUC of 0.83 (95% CI 0.80–0.86), whereas the PE-SCORE yielded an AUC of 0.78 on the development database. Although PE-SCORE had

lower prognostic performance than the logistic regression model, PE-SCORE performed similarly by AUC on both databases and offers real-world usefulness at the site-of-care.

It is plausible that definitions of candidate variables may be modified or optimized in future updates or revisions of prognostic tools. For example, other CT-derived variables, such as contrast reflux or the pulmonary arterial occlusion index, may be considered as candidate variables for a prognostic model. We only used CT RV:LV ratio from the CT. Initial oxygen saturation by pulse oximetry and initial respiratory rate at presentation were not retained. Both clinical variables were measured with patients at rest. Because exertional shortness of breath is a common symptom of PE, oxygen saturation and respiratory rate (measured after fixed and defined exertion) may yield different results when developing a prognostic tool. Even retained variables, like initial heart rate, can be optimized by measuring heart rate after fixed exertion or by using highest heart rate within a fixed time interval as a candidate variable.

Our prognostic model included creatinine elevation (greater than 2.0 mg/dL) as a parameter of renal function. Other reports have identified acute renal injury as a prognostic factor [70, 71]. Acute renal injury was not included as a candidate variable in PESI/sPESI. In our study, we did not attempt to differentiate renal injury from renal failure. We did not use glomerular filtration rate to assess renal function, and we used a modest cut-off for creatinine level evaluation for provider use. It is possible a different cut-off value of creatinine or a different renal function parameter of renal function assessment may offer an optimal prognostic performance.

Another possible limitation is that we used an ordinal GDE score of visually estimated severe RV dilatation (absolute or relative to left ventricle) and severe RV systolic function. Use of echocardiographic measurements on two-dimensional modality or other echocardiographic modalities may increase risk stratification stringency or provide recommendations for optimal cut-offs.

Although this study was performed at academic centers, competency in GDE has been expected of those emerging from EM residency training for the past decade. Our results may indicate an opportunity to study the impact of employing GDE into PE risk stratification. Upon external validation, any real-world application of PE-SCORE would include recommendation that technically difficult or uninterpretable GDE images limit full use of PE-SCORE. None of the other eight variables used to calculate PE-SCORE were missing during development. When faced with absent GDE scores, providers should use available clinical information, recognizing the worst case scenario (that GDE is abnormal) has not been ruled out. Providers may either add a point or consider the partial PE-SCORE a minimum score. The other real-world option is to consider comprehensive echocardiography (by cardiology service).

Most of the clinical outcomes were determined during hospitalization and may not have been recognized outside of the hospital setting. The study design did not directly assess the impact or safety of implementing the prognostic prediction or its PE-SCORE on provider decisions regarding disposition, level of monitoring needed, or escalation of treatment.

Potential benefits of PE-SCORE include early detection of deterioration and avoiding misclassification of patients who experience the outcome but would have been classified as low-risk by another prognostic tool. Potential harms may include unnecessary testing or interventions in those who did not experience any clinical deterioration outcomes despite higher risk classification, subjecting them to potential adverse events of the interventions, and increased lengths of stay and medical costs. After external validation, we anticipate use of the PE-SCORE tool in acute care settings with similar prevalence of early clinical deterioration to identify PE patients likely to benefit from early discharge and those who may need higher level monitoring and escalated PE interventions. However, incorporation of any new prognostic tool into

clinical practice requires implementation and impact studies to better understand the clinical consequences [72].

## Conclusions

We have summarized development and validation of a new prognostic tool that uses readily available imaging findings from CT, GDE, vital signs, and interview information. A PE-SCORE score of zero conferred a low probability and a score of $\geq 6$ predicted high probability of clinical deterioration/death within days of PE diagnosis. External validation may support use of this prognostic tool to inform decisions about early outpatient management versus the need for hospital-based monitoring and considerations for escalated PE interventions.

## Supporting information

**S1 Table. Univariable analysis of 138 candidate variables for primary outcome on development database.**
(DOCX)

**S2 Table. Comparison of clinical research sites in development database.**
(DOCX)

**S3 Table. Prognostic performance of sPESI and ESC at low-risk threshold.**
(DOCX)

**S1 Fig. Assignment of points to each of the nine variables in the PE-SCORE model.**
(DOCX)

**S1 Data.**
(CSV)

**S2 Data.**
(CSV)

**S3 Data.**
(CSV)

## Acknowledgments

Authors thank Pilar Tochiki for central database acquisition and management, Melanie Hogg for central site research project management, Kelly Goonan for scientific writing assistance, and Michael Runyon for mentoring.

## Author Contributions

**Conceptualization:** Anthony J. Weekes.

**Formal analysis:** Anthony J. Weekes, H. James Norton.

**Funding acquisition:** Anthony J. Weekes.

**Investigation:** Anthony J. Weekes, Jaron D. Raper, Kathryn Lupez, Alyssa M. Thomas, Carly A. Cox, Dasia Esener, Jeremy S. Boyd, Jason T. Nomura, Jillian Davison, Patrick M. Ockerse, Stephen Leech, Jakea Johnson, Eric Abrams, Kathleen Murphy, Christopher Kelly.

**Methodology:** Anthony J. Weekes, H. James Norton.

**Project administration:** Anthony J. Weekes.

**Validation:** Anthony J. Weekes.

**Visualization:** Anthony J. Weekes.

**Writing – original draft:** Anthony J. Weekes, Jaron D. Raper.

**Writing – review & editing:** Anthony J. Weekes, Jaron D. Raper, Kathryn Lupez, Alyssa M. Thomas, Carly A. Cox, Dasia Esener, Jeremy S. Boyd, Jason T. Nomura, Jillian Davison, Patrick M. Ockerse, Stephen Leech, Jakea Johnson, Eric Abrams, Kathleen Murphy, Christopher Kelly.

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
