## [Decision Letter · Decision Letter 0]

4 Aug 2021

PONE-D-21-19015

Development and validation of a prognostic tool: pulmonary embolism short-term clinical outcomes risk estimation (PE-SCORE)

PLOS ONE

Dear Dr. Weekes,

Thank you for submitting your manuscript to PLOS ONE. After careful consideration, we feel that it has merit but does not fully meet PLOS ONE’s publication criteria as it currently stands. Therefore, we invite you to submit a revised version of the manuscript that addresses the points raised during the review process.

I applaud the authors for undertaking this large and important task--the need for a tool to help stratify patients for short-term adverse outcomes in PE is sorely needed, as is incorporation of an acute measure of RV function on echocardiography.

In addition to the comments by the reviewers, please consider the following when preparing a revision.

-It appears that acute and chronic RV dysfunction were treated equally, but i am not certain this is appropriate, since the goal of the paper is to include PE-related RV dysfunction (RVD). For example, patients with RVD from pre-existing post-capillary pHTN may have outcomes different from those with pre-existing RVD from pre-capillary pHTN which may also be different from those with acute onset pHTN from a PE--is it the RVD that leads to the bad outcome or is it the underlying lung or heart disease?  Compounding this is the fact that size and location of PE are not reported, which potentially further distorts the relationship between the PE, RV function, and outcomes (particularly since PE severity is emphasized in the discussion). i would be interested in seeing an analysis of only those with acute RVD, as i suspect that patients with chronic RVD have a greater likelihood of poor outcomes at regardless of PE size.

-I wonder about the extent to which the presence of sepsis accounts for some of the adverse outcomes as opposed to the PE?

-line 467-469: this does not really make sense to me as the putative pathway for right heart dysfunction leading to adverse outcomes is via reduced left heart function from reduced right-sided output. in addition, i think this statement needs a reference. also, can you include the number of patients that had troponin and BNP levels drawn?

-while the NPV point estimate is quite high, the lower end of the CI may to too low for some providers to feel safe discharging these patients--can you discuss this?

-was there any measurement of inter-rater reliability or central adjudication of RV findings on echo?

-feasibility of GDE is really outside the scope of this paper and i would suggest removing this section from the discussion.

-overall, the statistical methods are quite robust. i would consider, however, moving some of the technical details to a supplement as i suspect that many readers who will find this paper most interesting will be clinicians for whom the statistical details might be overwhelming/distracting.

We look forward to receiving your revised manuscript.

Kind regards,

Robert Ehrman, MD, MS

Academic Editor

PLOS ONE

Journal Requirements:

Reviewers' comments:

Reviewer's Responses to Questions

**Comments to the Author**

1. Is the manuscript technically sound, and do the data support the conclusions?

Reviewer #1: Yes

Reviewer #2: Yes

Reviewer #3: Partly

2. Has the statistical analysis been performed appropriately and rigorously? 

Reviewer #1: Yes

Reviewer #2: Yes

Reviewer #3: Yes

3. Have the authors made all data underlying the findings in their manuscript fully available?

Reviewer #1: Yes

Reviewer #2: Yes

Reviewer #3: Yes

4. Is the manuscript presented in an intelligible fashion and written in standard English?

Reviewer #1: Yes

Reviewer #2: Yes

Reviewer #3: Yes

5. Review Comments to the Author

Reviewer #1: Interesting paper.Some issues, although methodological aspects are very strong and accurate.

1) Deterioration should be better defined in abstract and in paper

2) Abstract: CI for AUC should be added

3) methods: Authors stated that they performed sensitivity analysis for each candidate. This should be better specified

4) Data cleaning is not clear

.

Reviewer #2: Overall

This paper represents a valiant effort to create a new discriminating decision-making tool for patients with pulmonary embolism including RV echo findings as a variable. There is a need for PE management decision-making tools that consider this variable. Overall though, this tool does not appear to reliably perform at the high level of some of the existing decision-making tools and therefore its utility in practice is at this time limited. Perhaps if validated in a larger cohort of patients outside of one of the original participating institutions it might strengthen the argument for the utility of this decision-making tool but as it stands this tool’s performance appears mediocre. The concept of a risk stratifying PE tool is not novel as there are several other well validated decision-making tools in use for determining low vs high risk PE. PESI, HESTIA, ESC and sPESI consistently have very good, reliably reproducible negative predictive value in multiple validation cohorts and do not require users have any ultrasound training.

While it is unique (and likely important) to include POCUS RV abnormalities as a variable in the tool and there is some literature that supports including this variable in the determination of low vs high risk PE (https://academic.oup.com/eurheartj/article/40/11/902/5263773), inclusion of this variable further limits the generalizability of this tool as there are still a great number of EM trained physicians and midlevel providers who are not POCUS trained or are only minimally pocus trained. Were this tool to demonstrate exceptional discrimination with the addition of the RV echo findings variable there would be an argument for using it however, it underperforms and is more complex to use as compared to existing tools.

As it stands, this paper represents a good idea that needs further reliably reproducible results (particularly in terms of its negative predictive value for low risk PE) before it would be a novel and worthwhile contribution to the current body of evidence of risk stratifying patients with PE. The result differences between the database and validation groups in some key areas (differences in false negatives between the database and validation group and the underpredicting of the tool for the outcome of interest at all scores 0-9) make this tool at face value seem to perform unreliably. Where it may be somewhat novel (but still requires further validation) is as a risk stratification tool for short term outcomes specifically for patients with ESRD and a PE.

Abstract

Line 17: This statement is particularly misleading since the validation group had an 8% proportion of the primary outcome in the zero score group.

Intro

Well written. Identifies the need for a decision tool that considers RV echo findings. Supports the need for a decision tool that can assess risk of short term deterioration and not risk of 30 day deterioration.

Methods

Enrollment criteria appropriate.

This study is aided by the diversity of its multiple institutions.

Line 110-112, 121-124: Using the need for volume expansion or pressors for documented hypotension as a marker of deterioration from PE may be confounded by the fact that your score includes septic patients. Worsening sepsis may cause the same deterioration which in the model of this study may be incorrectly attributed to the PE when really the hypotension is from the infection.

Methods for selection of variables seems appropriate.

Line 169-177: This change is unfortunate and as you cover in your discussion, may be a confounder.

Sample size calculations are appropriate.

Blinding to the development of different databases is a strength.

Line 232: Explanation of LASSO is appreciated and helpful to the reader.

Overall, this section is too long and in places too complex. As written, some of it is difficult to understand due to the use of some statistics that are not frequently encountered (ex Line 283). May benefit from simpler explanation of why these tests were utilized. This comes up again in the data section with the use of F1 and Hosmer-Lemshow. Statistics chosen do seem appropriate for the data set and aim of the study.

Results

Hosmer-Lemeshow for the regression model development database and particularly the validation database indicates poor fit.

F1 score for the high-risk pool seems poor given that this group should be skewed to see more subjects with the primary outcome (I’d assume individuals with a high risk score should experience an increased number of poor outcomes).

Confidence interval for the PPV of the high-risk group is quite wide so while the reported PPV looks good the certainty that it is actually close to 90% is not great (according to that confidence interval it could be less than 80% which at that point, how much utility does this test have?).

It appears the tool significantly underpredicted the primary outcome at every score level in the validation group, particularly in the categorized low risk groups (0-4), which is concerning if this tool is being advertised to discriminate between patients that have a high likelihood of deterioration and require admission vs low likelihood of deterioration and suitability for outpatient management (unless you plan on advertising this tool as dichotomous (less than or equal to zero is low risk, anything else isn’t)).

While the NPV in the low-risk database group with the cut-off set to “0” looks great (NPV 97.9% (93.6 -99.5%)) it underperforms in the validation group (91.9%). Some of the existing PE tools consistently perform at a NPV >97% over multiple validation studies. The PPV of the high-risk PE threshold group appears to suffer similarly where it performs well in the validation group but suffers in the database group. Your attached supplement shows sPESI underperforming as compared to your tool but sPESI was not validated for use in patients with severe renal disease which your database includes, so it is a somewhat misleading comparison. ESC appears to outperform your tool in the both the low risk database and validation cohorts.

408-414: Though several tests indicate good calibration, your data shows this tool underpredicted the primary outcome at every score level.

Discussion

While the AUC values for this tool are fair to good, the AUC precision recall values are not which is reflected in the less than robust PPV values for this score at the high threshold.

Lines 462-470: This is definitely an interesting thought.

Line 483-485: Sums up why this score (if proven to be more reliable in the future) would be a clinical difference maker.

Line 492-494: This is exactly what I believe needs to happen to make this more publishable.

Line 505-506: Agree.

Line 548-550: Agree and this should be a separate publication from this dataset.

Conclusion

Line 577-579: Depends on what you are considering a cutoff for low or high probability. Having clinical deterioration occur within 5 days in 8% of the zero score individuals of the validation group seems uncomfortably high when many of our other decision tools set the threshold at 2% (which was seen in the database group).

Reviewer #3: I would like to start by saying that I am not an expert on the pathology in question, but the considerations

brought by the Authors seem to me to be well argued and with adequate bibliographic references.

The aim of the study is exposed in a very clear way.

I really appreciated the effort to maintain a high degree of rigor in the statistical analysis.

Overall, the study is interesting and well conducted. However, I believe that some methodological aspects

can be improved. I have provided a detailed list of suggestions and references in an attached file.

6. PLOS authors have the option to publish the peer review history of their article (what does this mean?). If published, this will include your full peer review and any attached files.

Reviewer #1: **Yes: **Fabrizio D'Ascenzo

Reviewer #2: No

Reviewer #3: No

---

## [Author Response · Author response to Decision Letter 0]

7 Sep 2021

All reviewer and editor comments have been responded to in the attached file labeled "Response to Reviewers."

---

## [Decision Letter · Decision Letter 1]

18 Oct 2021

PONE-D-21-19015R1Development and validation of a prognostic tool: pulmonary embolism short-term clinical outcomes risk estimation (PE-SCORE)PLOS ONE

Dear Dr. Weekes,

Thank you for submitting your manuscript to PLOS ONE. After careful consideration, we feel that it has merit but does not fully meet PLOS ONE’s publication criteria as it currently stands. Therefore, we invite you to submit a revised version of the manuscript that addresses the points raised during the review process.

We look forward to receiving your revised manuscript.

Kind regards,

Christophe Leroyer

Academic Editor

PLOS ONE

Journal Requirements:

Reviewers' comments:

Reviewer's Responses to Questions

**Comments to the Author**

1. If the authors have adequately addressed your comments raised in a previous round of review and you feel that this manuscript is now acceptable for publication, you may indicate that here to bypass the “Comments to the Author” section, enter your conflict of interest statement in the “Confidential to Editor” section, and submit your "Accept" recommendation.

Reviewer #2: All comments have been addressed

Reviewer #4: (No Response)

2. Is the manuscript technically sound, and do the data support the conclusions?

Reviewer #2: Yes

Reviewer #4: Yes

3. Has the statistical analysis been performed appropriately and rigorously? 

Reviewer #2: Yes

Reviewer #4: I Don't Know

4. Have the authors made all data underlying the findings in their manuscript fully available?

Reviewer #2: Yes

Reviewer #4: Yes

5. Is the manuscript presented in an intelligible fashion and written in standard English?

Reviewer #2: Yes

Reviewer #4: Yes

6. Review Comments to the Author

Reviewer #2: Overall, this manuscript has significantly improved since initial submission. While some of my initial concerns about the overall usefulness of this decision tool still stand, in its current iteration this manuscript provides better explanations of how the tool variables were derived and validated and provides increased clarity about the authors’ intent for how the tool is to be used. This manuscript makes a much better argument for why efforts should be made to externally validate (PE-SCORE) and evaluate its performance outside of the institutions in which it was created. The efforts put forth by the authors to address commentary and concerns in my initial review were exhaustive. At this point in time I am satisfied with the revisions to this manuscript and the authors’ responses to my questions. No further revisions are recommended at this time.

Reviewer #4: Comments to the authors :

The authors used data from a prospective registry in which 6 emergency centers in the USA participated, to propose a prognostic score validated secondarily in a second registry.

Their score is based on 9 items, and is claimed to be able to predict a population at low risk of deterioration compared to a population at higher risk of deterioration.

The desire to develop a score with a pragmatic interest (promoting safe outpatient treatment versus keeping patients who may benefit from more intensive treatment in hospital) is commendable and the authors should be warmly congratulated.

However, I have several comments which for the moment limit the acceptability of this work.

Major comments :

- Methods : the authors chose as primary endpoint an unusual composite endpoint compared to other studies(1–3). The authors would have to justify the construction of this criterion more strongly. For example if the objective is to individualize a population to make the risk of death, we could expect death and death from pulmonary embolism as the main endpoint, as was the case with the PESI score. Conversely, if the objective is to individualize a population with more risk of degradation despite well-conducted anticoagulant treatment, the authors must justify why they did not follow criteria such as the one proposed in the PEITHO trial, for example.

- Population: It is surprising to find so few elderly people in the population included in the study, even though the incidence of pulmonary embolism rises after 75 years and that advanced age is a major prognostic factor(4). Do you have a reason for this? Is it explained by a selection bias?

- Results : Many studies find renal failure as a factor of poor prognosis(5), in the short term(6). The authors must be able to discuss this strongly, in particular on the possibility that their high creatinine leaves may cause loss of sensitivity.

- Discussion : I think that the authors should explain more strongly how their score would make it possible to bring things not provided by the multitude of scores already proposed and those included in the international recommendations. Even if this is discussed by the authors, it is annoying not to find the advanced age and cancer, which are well known factors associated with a poor prognosis, as factors integrated in the model.

Minor comments:

- population : one of the limits is the difficulty to extrapolate the results to inpatients diagnosed with PE, which represents about a third of PE cases. Can the authors discussed a little bit ?

- Globally, the draft is of interest, and well written, but some of the information may be put in appendix, in order to shorten it.

1. Roy P, Penaloza A, Hugli O, Klok FA, Arnoux A, Elias A, et al. Triaging acute pulmonary embolism for home treatment by Hestia or simplified PESI criteria: the HOME-PE randomized trial. Eur Heart J [Internet]. 2021 Aug 7;1–13. Available from: https://academic.oup.com/eurheartj/advance-article/doi/10.1093/eurheartj/ehab373/6345003

2. Konstantinides S V, Meyer G, Becattini C, Bueno H, Geersing G-J, Harjola V-P, et al. 2019 ESC Guidelines for the diagnosis and management of acute pulmonary embolism developed in collaboration with the European Respiratory Society (ERS). Eur Heart J [Internet]. 2020 Jan 21;41(4):543–603. Available from: https://academic.oup.com/eurheartj/advance-article/doi/10.1093/eurheartj/ehz405/5556136

3. Meyer G, Vicaut E, Danays T, Agnelli G, Becattini C, Beyer-Westendorf J, et al. Fibrinolysis for patients with intermediate-risk pulmonary embolism. N Engl J Med [Internet]. 2014;370(15):1402–11. Available from: http://www.ncbi.nlm.nih.gov/pubmed/24716681

4. Delluc A, Tromeur C, Ven F Le, Gouillou M, Paleiron N, Bressollette L, et al. Current incidence of venous thromboembolism and comparison with 1998 : a community-based study in Western France. Thromb Haemost. 2016;3–10.

5. Murgier M, Bertoletti L, Darmon M, Zeni F, Valle R, Del Toro J, et al. Frequency and prognostic impact of acute kidney injury in patients with acute pulmonary embolism. Data from the RIETE registry. Int J Cardiol [Internet]. 2019;291:121–6. Available from: https://doi.org/10.1016/j.ijcard.2019.04.083

6. Chopard R, Jimenez D, Serzian G, Ecarnot F, Falvo N, Kalbacher E, et al. Renal dysfunction improves risk stratification and may call for a change in the management of intermediate- and high-risk acute pulmonary embolism: results from a multicenter cohort study with external validation. Crit Care [Internet]. 2021;25(1):57. Available from: http://www.ncbi.nlm.nih.gov/pubmed/33563311

7. PLOS authors have the option to publish the peer review history of their article (what does this mean?). If published, this will include your full peer review and any attached files.

Reviewer #2: No

Reviewer #4: No

---

## [Author Response · Author response to Decision Letter 1]

27 Oct 2021

Itemized responses are attached for both reviews. The most recent is dated Oct 2021.

---

## [Editor Report · Decision Letter 2]

2 Nov 2021

Development and validation of a prognostic tool: pulmonary embolism short-term clinical outcomes risk estimation (PE-SCORE)

PONE-D-21-19015R2

Dear Dr. Weekes,

We’re pleased to inform you that your manuscript has been judged scientifically suitable for publication and will be formally accepted for publication once it meets all outstanding technical requirements.

Kind regards,

Christophe Leroyer

Academic Editor

PLOS ONE

---

## [Editor Report · Acceptance letter]

9 Nov 2021

PONE-D-21-19015R2 

Development and validation of a prognostic tool: pulmonary embolism short-term clinical outcomes risk estimation (PE-SCORE) 

Dear Dr. Weekes:

I'm pleased to inform you that your manuscript has been deemed suitable for publication in PLOS ONE. Congratulations! Your manuscript is now with our production department. 

Kind regards, 

on behalf of

Dr. Christophe Leroyer 

Academic Editor

PLOS ONE